# Study on the Effect of Different Viscosity Reducers on Viscosity Reduction and Emulsification with Daqing Crude Oil

**DOI:** 10.3390/molecules28031399

**Published:** 2023-02-01

**Authors:** Fan Zhang, Qun Zhang, Zhaohui Zhou, Lingling Sun, Yawen Zhou

**Affiliations:** 1State Key Laboratory of Enhanced Oil Recovery, Research Institute of Petroleum Exploration & Development, Beijing 100083, China; 2School of Light Industry, Beijing Technology and Business University, Beijing 100048, China

**Keywords:** emulsification, viscosity reduction, water separation rate

## Abstract

The urgent problem to be solved in heavy oil exploitation is to reduce viscosity and improve fluidity. Emulsification and viscosity reduction technology has been paid more and more attention and its developments applied. This paper studied the viscosity reduction performance of three types of viscosity reducers and obtained good results. The viscosity reduction rate, interfacial tension, and emulsification performance of three types of viscosity reducers including anionic sulfonate, non-ionic (polyether and amine oxide), and amphoteric betaine were compared with Daqing crude oil. The results showed that the viscosity reduction rate of petroleum sulfonate and betaine was 75–85%. The viscosity reduction rate increased as viscosity reducer concentration increased. An increase in the oil–water ratio and polymer decreased viscosity reduction. When the concentration of erucamide oxide was 0.2%, the ultra-low interfacial tension was 4.41 × 10^−3^ mN/m. When the oil–water ratio was 1:1, the maximum water separation rates of five viscosity reducers were different. With an increase in the oil–water ratio, the emulsion changed from *o*/*w* emulsion to *w*/*o* emulsion, and the stability was better. Erucamide oxide and erucic betaine had good viscosity reduction and emulsification effects on Daqing crude oil. This work can enrich knowledge of the viscosity reduction of heavy oil systems with low relative viscosity and enrich the application of viscosity reducer varieties.

## 1. Introduction

After the primary and secondary production stages of extraction, 40% of an oil deposit remains in the stratum [1]. With the increase in oil demand and the decrease in conventional oil production, the development of heavy oil has been paid more and more attention by scholars. Globally, recoverable resources of heavy oil account for 29% of total unconventional oil and gas resources [2]. However, due to the complex stratum, low stratum temperature, high content of crude oil gel, and asphaltene, the stranded crude oil has high viscosity, which brings great challenges to its exploitation and transportation [3,4,5]. Therefore, reducing the viscosity of heavy oil and improving its fluidity are urgent problems to be solved [6,7,8,9,10].

Heavy oil viscosity reduction technology can be divided into physical viscosity reduction and chemical viscosity reduction. Physical viscosity reduction technologies include thermal energy viscosity reduction mining, dilution viscosity reduction mining, and pipeline heat and viscosity reduction transportation. Chemical viscosity reduction technologies include chemical viscosity reduction mining (emulsion viscosity reduction and oil soluble viscosity reduction, etc.), microbial viscosity reduction mining, and chemical viscosity reduction pipeline transport. Many oil reservoirs cannot be economically exploited by means of steam stimulation or electric heating because of dispersed blocks, small oil-bearing areas, and thin reservoirs. When oil pipelines are laid in deserts and under the sea, the traditional heating transportation method cannot adapt to the harsh environmental requirements. Chemical viscosity reduction is a mining method that adds a certain proportion of chemical additives to heavy oil to reduce its freezing point and flow viscosity and restrain paraffin precipitation in the process of oil pumping and transportation. The chemical viscosity reduction method is widely used; the system is easy to adjust, more selective, and has a high technical and economic value. Therefore, chemical viscosity reduction technology is more important for the exploitation and transportation of heavy oil [11,12].

Emulsifying viscosity reduction technology has the characteristics of a remarkable viscosity reduction effect, easy adjustment of the system, wide selectivity, and high technical and economic value. For example, the application of downhole emulsification viscosity reduction technology can improve pump efficiency and the fluid level of the well, reduce energy consumption, and increase the oil production of a single well. Therefore, emulsification viscosity reduction has been widely studied and applied in research and actual production. Some emulsification viscosity reduction can even reduce the viscosity more than 99%. According to their chemical structure, emulsifiers are usually divided into cationic, anionic [13,14,15], nonionic, amphoteric, and nonionic—anionic complex. However, cationic viscosity reducers are easily adsorbed or precipitated by the formation, so they are rarely used as an oil displacement agent or emulsified viscosity reducer. At present, anionic viscosity reducers are mainly studied. For example, 0.1 wt% fatty alcohol polyoxyethylene ether sulfate (SC) and 0.05 wt% xanthan gum (XG) were combined to develop a novel surfactant–polymer system. The results showed that the addition of XG into SC systems could obviously decrease the viscosity ratio of oil to displaced water to 1.42 and simultaneously maintain the high viscosity reduction rate at 94.03%, which was beneficial for decreasing the mobility ratio of water to oil [16]. The viscosity reduction performance and the effect of different chain lengths on the viscosity reduction effect were also investigated. Viscosity reduction, emulsification, wetting, and foaming performance tests showed that the viscosity reduction performance of this series of polymeric surfactants was excellent, with a viscosity reduction rate exceeding 95% [13]. There are few studies on nonionic and amphoteric viscosity reducers.

Therefore, in this paper, anionic, nonionic, and amphoteric viscosity reducers were selected to reduce the viscosity of Daqing crude oil in order to evaluate viscosity reduction performance before oil flooding, including the viscosity reduction rate, interfacial tension, and water separation rate. The viscosity reduction performance differences of different types of viscosity reducers were obtained using the investigation of viscosity reducer concentration, oil–water ratio, alkali and polymer factors, and the viscosity reducer system with good viscosity reduction and emulsification effect for Daqing crude oil were obtained. This work can enrich knowledge of the viscosity reduction of heavy oil systems with low relative viscosity and enrich the application of viscosity reducer varieties.

## 2. Results and Discussion

### 2.1. Viscosity Reduction Rate

Emulsifying viscosity reduction is an important technical means to enhance oil recovery. By injecting aqueous viscosity reducer solution into the formation, the special structure of the viscosity reducer can emulsify the oil in the formation and change high-viscosity crude oil into a low-viscosity W/O emulsion. In this way, the adsorption capacity of crude oil in the formation decreases and oil recovery improves. Therefore, the viscosity reduction rate is an important evaluation method for the oil displacement performance of the viscosity reducer. The higher the viscosity reduction rate, the lower the viscosity of the system and the more beneficial to enhanced oil recovery.

(1)The Effect of Viscosity Reducer Type and Concentration on Viscosity Reduction Rate

Generally, the higher the viscosity reducer concentration, the higher the viscosity reduction rate; however, the corresponding oil-recovery cost is also higher, so it is very important to choose an appropriate viscosity reducer concentration. The viscosity of the emulsion was measured at 45 °C with the mass concentration of viscosity reducer at 0.3% and the viscosity reduction rate was calculated. At the same time, the viscosity and viscosity reduction rate of the system with a mass concentration range of 0.05 ~ 0.5% were measured with petroleum sulfonate (2#); the results are shown in Table 1.

As can be seen from Table 1, with the increase in the mass concentration of 2# viscosity reducer, the viscosity of the emulsion decreased and the viscosity reduction rate increased. This is because petroleum sulfonate itself is a classic viscosity reducer, and studies have shown that it has good emulsifying properties [17,18,19,20]. When the mass concentration of the 2# reached 0.5%, the viscosity of the emulsion further reduced, but there was little difference in the viscosity when the mass concentration was 0.3%. Considering the cost, a practical application would be more inclined to choose 0.3% mass concentration. At the same time, comparing the viscosity of 1#, 2#, 3#, 4#, and 5# at a mass concentration of 0.3%, 5# emulsion had the lowest viscosity, the highest viscosity reduction rate and the best effect. This indicated that when the oil–water ratio was 5:5, betaine amphoteric viscosity reducer had a very effective viscosity reduction effect on the crude oil system, and for 4#, although it was also a sulfonate surfactant, the emulsification effect with Daqing crude oil was very poor, so the viscosity reduction rate was very low. For 1# and 3# non-ionic surfactants, the viscosity reduction rate was moderate.

The reason for the viscosity reduction of viscosity reducers is that their molecular structure contains polar functional groups, which have strong permeability and hydrogen bonding ability. Therefore, they are dispersed between the colloidal and asphaltene lamellar molecules of crude oil, which releases the liquid oil wrapped in the micelle structure and increases the dispersion of the heavy oil system. At the same time, the hydrophobic alkyl long chain in the molecules of the viscosity reducer can fully extend around the asphaltene aggregate to form a solvent layer, which plays a good shielding role on the oil droplets. In this manner, the viscosity reducer reduces the viscosity of a heavy oil emulsion. It can be seen from the results that the betaine amphoteric viscosity reducer had a strong ability to disperse crude oil and the best viscosity reduction effect because its molecular structure contained anionic and cationic polar functional groups and its hydrophobic group played a good solvent shielding role. That the viscosity of the emulsion decreased with an increase in viscosity reducer concentration may be because when the viscosity reducer concentration was low, it had a low-strength oil–water interface layer and it was easy for the droplets to coalesce and form an emulsion with poor stability, making the viscosity of the emulsion relatively high. With an increase in viscosity reducer concentration, its arrangement at the interface became closer, the strength of the interface layer increased, and the droplets did not easily coalesce. The viscosity reducer had a strong ability to reduce the oil–water interface and formed a stable emulsion. Therefore, the viscosity of the emulsion decreased a great deal.

(2)The Effect of Oil–Water Ratio on Viscosity Reduction Rate

The emulsion viscosity and viscosity reduction rate of the 2# at three concentrations were measured with different oil–water ratios (7:3, 5:5, and 3:7). The measurement results are shown in Table 2.

It can be seen from Table 2 that viscosity decreased and viscosity reduction rate increased with an increase in the concentration of viscosity reducer at the three oil–water ratios. When the oil–water ratio was 3:7, the lotions were oil-in-water emulsion due to containing more water than oil and the effect of surfactant. The viscosity of the three emulsions were lower than 2 mPa·s and the viscosity reduction rate was more than 90% compared with 20.3 mPa·s for Daqing crude oil. When the oil–water ratio was 5:5, the three lotions were also oil-in-water emulsions, but due to the increase in the oil ratio, the viscosity of the lotions were between 4–6 mPa·s, which was significantly higher than those of the systems with an oil–water ratio of 3:7; the viscosity reduction rate was between 70–80%. Therefore, the viscosities for the 3:7 oil–water ratio were lower than those for the oil–water ratio of 5:5, so the viscosity reduction rate was better. After calculation using Formula 1, the viscosity reduction rate was more than 90%, which has a better viscosity reduction effect on crude oil. Compared with the viscosity reduction rate of the three oil–water ratios with the same mass concentration, it can be seen that viscosity increased with the increase in oil content in the system. When the ratio of oil and water was 7:3, the viscosity of the emulsion exceeded the viscosity of the crude oil itself and the viscosity reduction rate also became negative. The reason was that the oil content of the system was too high, and the emulsion changed from an oil-in-water to a water-in-oil system. As a result, the droplet shape at the water–oil interface changed, the particle size increased and the viscosity increased, but the viscosity exceeded the viscosity of crude oil.

(3)The Effect of Polymer on Viscosity Reduction Rate

The viscosity and viscosity reduction rate of the emulsion were determined at a concentration of 0.3% and the volume ratio of oil and water was 1:1 at different polymer concentrations (0, 1000 ppm, and 1500 ppm). The experimental results are shown in Table 3.

Polymer flooding is designed to enhance oil recovery by injecting water-soluble polymers to increase the viscosity of the water phase and the sweep volume [21]. Therefore, polymers are often used to change the viscosity of the system. As can be seen from Table 3, the viscosity of the system without the polymer was 4.7 mPa·s and the viscosity reduction rate was 76.85%. However, the viscosity of the system increased after the addition of the polymer and the viscosity was greater than that of the polymer solution. This is because the viscosity of the polymer itself leads to an increase in the viscosity of the emulsion. Second, the interaction between viscosity reducer molecules and polymer molecules promoted the extension of viscosity reducer and polymer molecular chains and enhanced the entanglement between them, thus leading to the increase in viscosity.

### 2.2. Interfacial Tension

(1)The Effect of Viscosity Reducer Type and Concentration on Interfacial Tension

As shown in the results in Figure 1a–c, the interfacial tension of the three viscosity reducer systems quickly reached equilibrium within 5 min when the viscosity reducer concentration was 0.05–0.5% and then the interfacial tension value did not change much; the equilibrium interfacial tension first decreased and then increased with the increase in viscosity reducer concentration. The lowest interfacial tension value was reached when the concentration was 0.2%. Comparing the three kinds of viscosity reducers, the lowest interfacial tension values were in the order of 3# > 1# > 2# and the lowest interfacial tension of the 3# viscosity reducer system reached the ultra-low interfacial tension of 4.41 × 10^−3^ mN/m. The minimum interfacial tension of mN/m for the 1# viscosity reducer system was 10^−2^ mN/m and the lowest interfacial tension for the 2# viscosity reducer system was between 1–10 mN/m.

According to the results in Figure 1d, the interfacial tension of the 5# viscosity reducer system stabilized and reached ultra-low interfacial tension (≤10^−3^ mN/m) after 105 min, while other viscosity reducer systems reached equilibrium interfacial tension quickly. The lowest interfacial tension for 2# and 4# sulfonates was larger than 1 mN/m, which indicated that the two sulfonates had poor interfacial tension reduction ability for the crude oil in the Daqing oilfield. As for the 3# nonionic viscosity reducer, the interfacial tension rapidly reached its ultra-low level within 5 min, but with increasing time the interfacial tension increased slightly and finally stabilized at 10^−2^ mN/m. It can be seen that, due to its hydrophilic and lipophilic amphiphilic structure, the viscosity reducers formed an interfacial adsorption layer with great strength at the oil–water interface, especially 3# and 5#, so they could better reduce the interfacial tension.

(2)The Effect of Alkali on Interfacial Tension

As can be seen in Figure 2, ultra-low interfacial tension was not attained by viscosity reducer 2# without Na_2_CO_3_. When 1.2% Na_2_CO_3_ was added to the system, the interfacial tension of the system greatly decreased to 4.60 × 10^−3^ mN/m at 25 min and was kept in the ultra-low interfacial tension range the whole time.

The reason that alkali significantly decreased interfacial tension was not only because electrolyte salts affect the adsorption of the viscosity reducer and the synergistic effect between alkali and the viscosity reducer but also because alkali reacts with the organic acids in crude oil to form an organic viscosity reducer [22]; it had an obvious synergistic effect with petroleum sulfonate [23]. In general, the addition of alkali can greatly decrease interfacial tension, but the alkali itself damaged the formation, so it was also very important to choose the appropriate alkali concentration.

### 2.3. Water Separation Rate

(1)The Effect of Viscosity Reducer Type on Water Separation Rate

For testing, the ratio of oil to water was 1:1, and the mass concentration of viscosity reducer was 0.3%. The water separation rates of five viscosity reducers were measured, and the results are shown in Figure 3.

The water separation rate is an important parameter for characterizing emulsion stability. The lower the water separation rate, the more stable the emulsion. Figure 3 shows that under the same experimental conditions, all five viscosity reducers reached the water separation equilibrium time within 1 day but the maximum water separation rate was very different. The order of the maximum water separation rate f_max_ was 2# > 4# > 5# > 1# > 3#.

The water separation time for the 2# viscosity reducer was the shortest. The water separation rate of the emulsion reached 92% within 60 min; the maximum water separation rate was 98% within 48 h and the emulsion was basically demulsified completely. The maximum water separation rate (fmax) of the 3# viscosity reducer was 72% and the water separation rate was the lowest, indicating that the emulsion formed with crude oil had the best stability. The interfacial tension formed by the 3# viscosity reducer with crude oil was also the lowest, so the low interfacial tension was conducive to the emulsion [24]. The reason is that the viscosity reducer forms a stable interface layer at the oil–water interface, which reduces the oil–water interfacial tension. In this way, the oil droplets are dispersed and prevented from coalescing. Therefore, the interfacial tension is directly related to the stability of the emulsion.

(2)The Effect of Oil–Water Ratio on Water Separation Rate

It can be seen from the results in Figure 4 that, with an increase in the oil–water ratio, the water fraction of the emulsion gradually decreased. The f_max_ of an oil–water ratio of 3:7 was 91% and the f_max_ of an oil–water ratio of 5:5 was 72%, both of which were much greater than that of 6.6% when the oil–water ratio was 7:3. This is because when the amount of water is substantially more than oil, an O/W emulsion can easily be formed and the viscosity of the solution is decreased. When the oil–water ratio is greater than 1:1, that is, when the amount of oil in the system is greater than the water, it is easier for a W/O emulsion to form. Because the viscosity of crude oil is greater than that of the water phase and the stability of the emulsion is affected by the oil–water interface film, the strength of the droplet interface film of the W/O emulsion was more stable and the dispersed phase could not easily coalesce. The macro performance of an emulsion with such a high oil–water ratio is not easy to demulsify and the emulsion is more stable, so the water separation rate is lower. However, in actual application, a high oil–water ratio will lead to an increase in viscosity, which is not conducive to actual crude oil extraction, so industrial applications often choose a lower oil–water ratio.

Figure 5 shows the emulsion state of the 3# viscosity reducer at 48 h. From left to right, the oil–water ratio was 3:7, 5:5, and 7:3. It can be seen that the volume of *o*/*w* emulsion gradually decreased with the increase of the oil–water ratio and the emulsion changed from *o*/*w* to *w*/*o*. In particular, when the oil–water ratio was 7:3 the solution was a very stable *w*/*o* emulsion.

### 2.4. Microstructure

The size and state of the emulsion droplets can be directly observed using a biologic microscope. Generally speaking, well-distributed droplets with smaller sizes indicate a better emulsifying ability of the viscosity reducer. The effects of microstructure can further explain the rationality of viscosity reduction.

As can be seen in Figure 6, at 10 min, an obvious O/W emulsion was formed with a mass concentration of 0.3% by the 3# viscosity reducer when the oil–water ratio was 3:7 and 5:5; the droplet diameters of the emulsion had little difference. However, emulsion drops in an oil–water ratio of 5:5 had better dispersion and more uniform distribution. The overall state of the emulsion with the two oil–water ratios did not change much, but the diameter of the droplets slightly increased with an increase in time. The droplet particles became significantly larger and there was an obvious coalescence phenomenon. Demulsification started at about at 30 min and the water separation rate reached about 20% at 2 h, which corresponds to the water separation rate research results. At an oil–water ratio of 7:3, the emulsion formed an obvious W/O structure and the emulsion was stable. After 2 h, there was still no significant change, which was consistent with the water separation rate and viscosity reduction rate of the macro lotion state.

## 3. Materials and Methods

### 3.1. Reagents and Instruments

The crude oil used in the experiment was a block of crude oil from Daqing, China, with a viscosity of 20.3 mPa·s (45 °C, 0.101 MPa) and a density of 0.85 g/cm^3^ after dehydration (45 °C, 0.101 MPa). There were five viscosity reducers used in the experiment: carboxylate polyoxyethylene ether (1#), petroleum sulfonate (2#), erucamide oxide (3#), alkyl xylene sulfonate (4#), and erucic betaine (5#). These viscosity reducers were made in the laboratory and used after purification. Sodium carbonate, sodium bicarbonate, sodium sulfate, potassium chloride, sodium chloride hexahydrate magnesium chloride, calcium chloride, and polyacrylamide were all analytically pure and obtained from Beijing Chemical Plant. The main instruments required for the experiment were a biofluorescence microscope (Leica DM3000 LED, Wetzlar, Germany), rotary dropper interface tensimeter (TX-500C, USA KINO Industry Co., Boston, MA, USA), and Brookfield viscometer (LVDV-IIIU, Brookfield, MA, USA). The experimental water was simulated formation water, and the ion composition is shown in Table 4.

### 3.2. Experimental Method

#### 3.2.1. Viscosity Reduction Rate

Viscosity reduction rate is the difference between the viscosities of a viscosity reducer emulsion and crude oil and the percentage of the viscosity of crude oil. Emulsion viscosity was measured using a Brookfield LVDV-IIIU viscometer. The mixtures of crude oil and viscosity reducer were added to 25 mL bottles according to different oil–water ratios. After being emulsified with a homogenizer at a speed of 10,000 r/min for 1 min, 15 mL of emulsion was added to the viscometer. The viscosity was measured at a speed of 6 rpm and the measurement results were recorded to calculate the viscosity reduction rate. The viscosity reduction rate formula is as follows:y = (η_oil_ – η_em_)/η_oil_ × 100%(1)
where y is the viscosity reduction rate (%); η_oil_ is the viscosity of crude oil in mPa·s; and η_em_ is the viscosity of the emulsion sample.

#### 3.2.2. Interfacial Tension

A TX-500C interfacial tensimeter was used to measure interfacial tension. The sample tube was successively cleaned with petroleum ether, acetone, and deionized water and the sample to be tested was moistened. When loading the sample, the continuous phase was slowly injected into the sample tube until the continuous phase had filled the sample tube. The oil phase of about 1.0 μL was injected into the continuous phase of the sample tube with a microsyringe. The lid of the sample tube was tightly closed and there was no bubble in the sample tube. After preheating, the rotating speed of the test software was adjusted to the set speed of 5000 r/min. The calculation formula of interfacial tension was
σ = 1.2336 × π^2^ · ∆p × 10^6^·(d^3^/m^3^)·(1/n^3^)·(1/R^2^)·f(L/d)(2)
where σ is the interfacial tension, mN/m; ∆p is the density difference of the two phases, g/cm^3^; R is the instrument panel speed reading, ms/r; n is the external refractive index; d is the oil drop width, cm; L is the droplet length, cm; M is the instrument microscope magnification; and f(L/d) is the correction factor. When L/d ≥ 4, f(L/d) is equal to 1.

#### 3.2.3. Water Separation Rate

The crude oil and viscosity reducer were mixed in a small bottle according to a volume ratio of 1:1 and emulsified for 1 min with a homogenizer at a rotation speed of 10,000 rpm/min. Then, emulsion was poured into the measuring cylinder and placed in the thermostat at the temperature of the reservoir. The volume of each phase was recorded and the water separation rate was calculated. The formula of the water separation rate is as follows:f = V/V_0_ × 100%(3)
where f is the water separation rate, V is the volume of precipitated water in mL; and V_0_ is the volume of viscosity reducer solution in mL.

#### 3.2.4. Microstructure

The unstable demulsification process of the emulsion was observed using a bioluminescence microscope. A total of 2 μL of emulsion was immediately placed on a glass sheet, and the microscopic state of the emulsion was observed after 10 min, 30 min, and 2 h under a bioluminescence microscope at 200 times magnification.

## 4. Conclusions

In summary, we studied the viscosity reduction rate, interfacial tension, and emulsification performance of anionic, nonionic, and amphoteric viscosity reducers with influencing factors such as the type of viscosity reducer, mass concentration, oil–water ratio, polymer, and alkali. The results showed that anionic petroleum sulfonate and amphoteric betaine had good viscosity reduction rates. An increase in oil–water ratio and polymer increased the viscosity of the emulsion and led to a decrease in viscosity reduction. For anionic and non-ionic viscosity reducers, the interfacial tension first decreased and then increased with an increase in viscosity reducer concentration. The ultra-low interfacial tension reached 4.41 × 10^−3^ mN/m when the concentration of 3# viscosity reducer was 0.2%. However, the 2# viscosity reducer system reached ultra-low interfacial tension with added 1.2% Na_2_CO_3_. When the oil–water ratio was 1:1, the order of the maximum water-separation ratios of the five viscosity reducers was 2# > 4# > 5# > 1# > 3#. For the 3# viscosity reducer system, a high oil–water ratio formed a W/O emulsion and a low water-separation ratio. However, the system with a low oil–water ratio was an O/W emulsion and had a better viscosity reduction effect. In conclusion, the nonionic and betaine viscosity reducers had good viscosity reduction emulsification effects on Daqing crude oil.

## Figures and Tables

**Figure 1 molecules-28-01399-f001:**
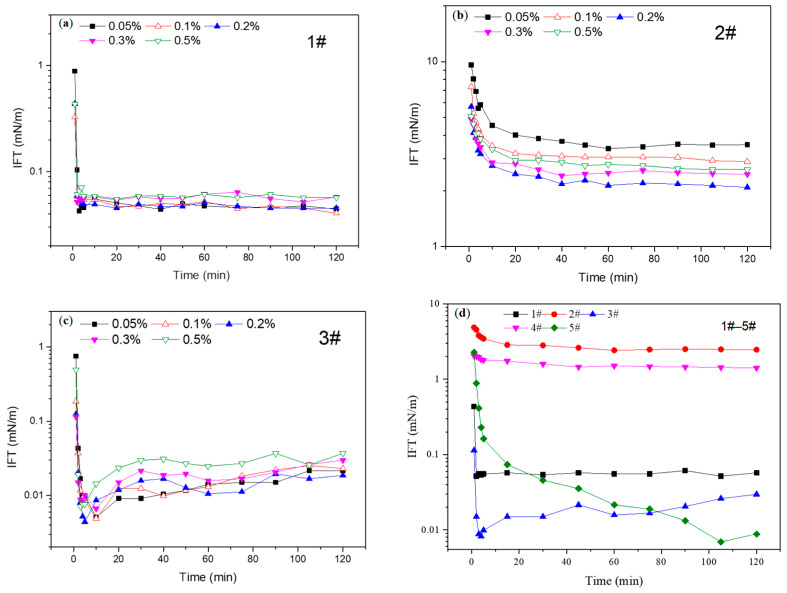
Interfacial tension of different viscosity reducers. (**a**) 1#, (**b**) 2#, (**c**) 3#, (**d**) 1#–5# with a concentration of 0.3%.

**Figure 2 molecules-28-01399-f002:**
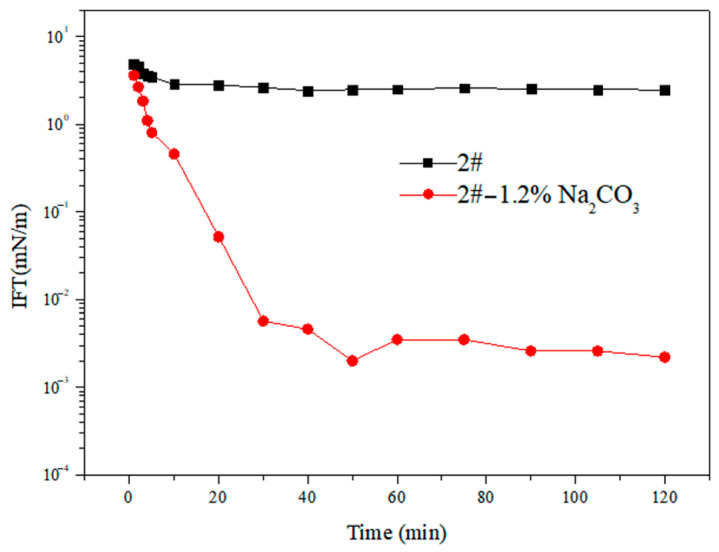
Effect of alkali on interfacial tension over time.

**Figure 3 molecules-28-01399-f003:**
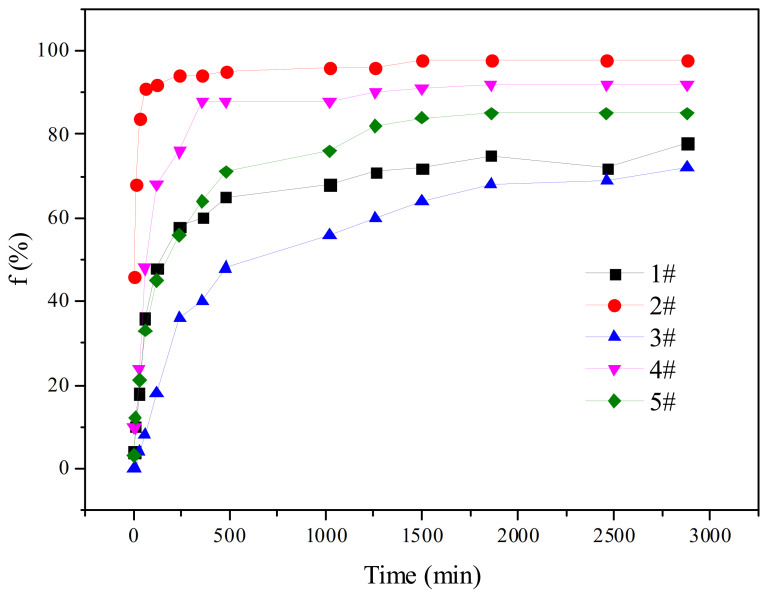
The water separation rates of different viscosity reducer systems over time.

**Figure 4 molecules-28-01399-f004:**
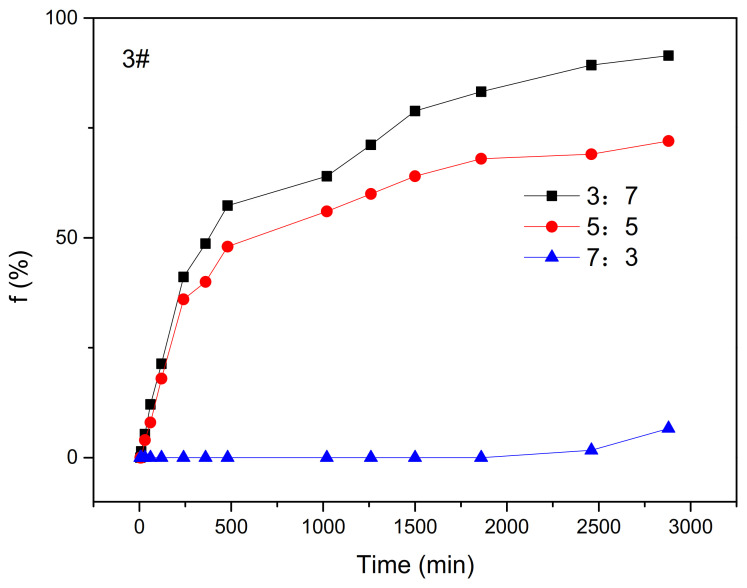
The water separation rate of different oil–water ratios over time.

**Figure 5 molecules-28-01399-f005:**
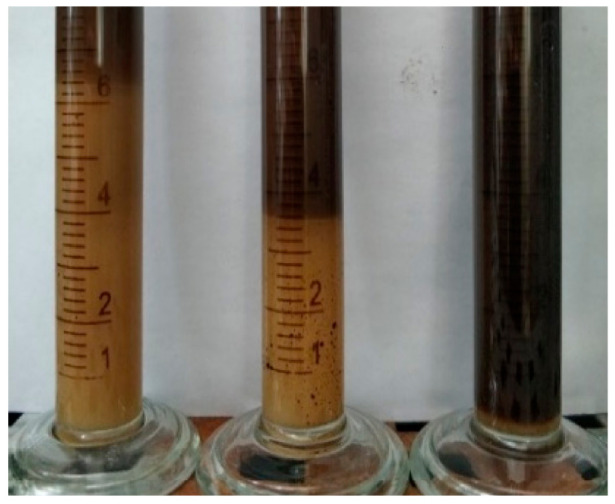
Emulsion states at different oil–water ratios at 48 h.

**Figure 6 molecules-28-01399-f006:**
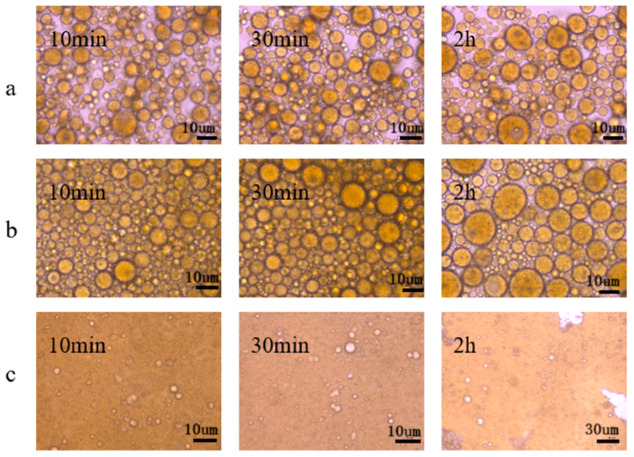
Microstructure of emulsions at different oil–water ratios. (**a**) 3:7, (**b**) 5:5, (**c**) 7:3.

**Table 1 molecules-28-01399-t001:** Viscosity reduction results.

Viscosity Reducer	Viscosity/mPa·s	Mass Concentration/%	Viscosity Reduction Rate/%
petroleum sulfonate (2#)	7.8	0.05	61.57
petroleum sulfonate (2#)	5.6	0.1	72.41
petroleum sulfonate (2#)	5.4	0.2	73.40
petroleum sulfonate (2#)	4.7	0.3	76.85
petroleum sulfonate (2#)	4.6	0.5	77.34
carboxylate polyoxyethylene ether (1#)	11.5	0.3	43.35
erucamide oxide (3#)	7.7	0.3	62.07
alkyl xylene sulfonate (4#)	17.3	0.3	14.78
erucic betaine (5#)	3.2	0.3	82.24

**Table 2 molecules-28-01399-t002:** Viscosity reduction results for 2# with different oil–water ratios.

Oil–Water Ratio	Mass Concentration/%	Viscosity/mPa·s	Viscosity Reduction Rate/%
3:7	0.1	1.9	90.64
	0.3	1.8	91.13
	0.5	1.6	92.12
5:5	0.1	5.6	72.41
	0.3	4.7	76.85
	0.5	4.6	77.34
7:3	0.1	51.9	−155.67
	0.3	46.7	−130.05
	0.5	43.2	−112.81

**Table 3 molecules-28-01399-t003:** Viscosity reduction results with different polymer concentrations.

Polymer/ppm	0	1000	1500
Viscosity/mPa·s	4.7	30.1	59.1
Viscosity reduction rate/%	76.85	−48.28	−191.13

**Table 4 molecules-28-01399-t004:** Ionic composition of simulated formation water.

Ion	CaCl_2_	MgCl_2·_6 H_2_O	NaCl	NaHCO_3_	KCl	Na_2_SO_4_	Na_2_CO_3_	Total Content
Content/g·L^−1^	0.0282	0.0269	1.3066	1.3412	5.5907	0.0339	0.7995	7887.65

## Data Availability

The data presented in this study are available on request from the corresponding author.

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
