# Peer review of "Study on the Effect of Different Viscosity Reducers on Viscosity Reduction and Emulsification with Daqing Crude Oil"

_molecules, 2023, doi:10.3390/molecules28031399_

Round 1
Reviewer 1 Report
Thank you for the opportunity to review this manuscript. Overall, the study investigates the effectiveness of various viscosity reducers in improving the fluidity of heavy oils.
There are a few concerns that need to be addressed before the manuscript can be accepted for publication. Firstly, the order of the chapters should be corrected, with the methodology section coming before the results. Secondly, the term "reduce viscosity rate" is not clearly defined and requires further explanation. Thirdly, the results presented in Table 3 are somewhat confusing, as it is not clear how viscosity can be reduced by more than 190%. Finally, the data in Table 2 needs to be more thoroughly discussed, as the 90% reduction in viscosity seems to imply that the emulsion has a viscosity of only 10cP, which is not explained in the manuscript. Additionally, the "relative" reduction in viscosity should be compared only to the effect of the polymer on the emulsion, rather than the presence or absence of the viscosity reducer.
Overall, while the study investigates an important topic, the manuscript needs further work to clarify the methodology and results.
Author Response
According to the reviewer's comments, we replied to the reviewer's comments and revised the paper in detail. See word for details.

Reviewer 2 Report
In this paper, the viscosity reduction rate, interfacial tension and emulsification properties of anionic, nonionic and amphoteric viscosity reducer with Daqing crude oil were studied. There are some interesting findings regarding the surfactants in this work, however, the manuscript is more like a lab report instead of an academic paper. The manuscript must have a major revision to make it more academic and in-depth.
1. In the abstract, it is important to highlight the detailed outcomes of your paper and the experimental methods that have been done in this paper, how it looks like for your test rigs.
2. For the literature review. You need to provide critical analysis and discussion. It is important to provide a more detailed literature review. And also, highlight why your project is important and what is novel in detail.
3. I do not think the paper have a good logic, it is better to put the 'results and discussion' section at the end of the 'materials and methods'
4. The experimental work were conducted using the Daqing crude oil. Whether the findings of this work can be applied to other types of oil is not fully discussed by the authors. based on this, the reviewer does not think the title is suitable for the work of this paper. this issue is my primary concern.
5. some typo and grammar errors should be avoided, such as Line 201 fmax
Author Response

(The authors gave the same response as above.)

Round 2
Reviewer 1 Report
The paper can be accepted.
Author Response
Thank you very much!!!
Reviewer 2 Report
It is suggested that more theoretical details should be included for the mechanism about the effect of viscosity reducer on viscosity reduction and emulsification.
Author Response

(The authors gave the same response as above.)
